# Study on the Mechanism of the Pink Tooth Phenomenon Using Bovine Teeth: A Pilot Study

**DOI:** 10.3390/diagnostics13162699

**Published:** 2023-08-17

**Authors:** Nozomi Sumi, Saki Minegishi, Jun Ohta, Hajime Utsuno, Koichi Sakurada

**Affiliations:** Department of Forensic Dentistry, Tokyo Medical and Dental University, 1-5-45 Yushima, Bunkyo-ku, Tokyo 113-8510, Japan; saki-m.fde@tmd.ac.jp (S.M.); kppscl.lms.lcn@gmail.com (J.O.); hazimeu.fde@tmd.ac.jp (H.U.)

**Keywords:** pink teeth, bovine teeth, L*C*h color space, carboxyhemoglobin, *Serratia marcescens*

## Abstract

The pink teeth phenomenon has occasionally been observed in forensic autopsies. This study aimed to establish an experimental pink tooth model and an objective color tone evaluation method in order to clarify changes in the color tone of teeth and the relationship with hemoglobin monoxide and its decomposition products and with red pigment-producing bacteria, under various external environmental factors. It was confirmed that the color tone evaluation with ΔE and the L*C*h color space was useful. The results of various examinations using this model showed that color development was suppressed under aerobic conditions, faded early under light, became bright red under a low temperature and showed a tendency to be reddish at 3 days under high humidity and in the presence of soft tissue. The biochemical analysis revealed a significant increase in carboxyhemoglobin at 7 days and a tendency toward increasing the total heme pigment and bilirubin levels over time. The bacteriological analysis revealed that red pigment-producing bacteria increased over time but that the color faded after 7 days. These results suggest that putrefaction greatly affects the pink teeth phenomenon, whereas red pigment-producing bacteria have little effect on the occurrence of pink teeth. However, further studies are needed to clarify bacteriological involvement.

## 1. Introduction

The pink teeth phenomenon, in which the teeth of cadavers have a pink hue, is occasionally observed in forensic autopsies. This phenomenon was first described in 1829 by Thomas Bell [1], who noted that teeth turned pink in persons who died by hanging or drowning. Borrman et al. [2] noted that little attention was paid to this phenomenon, and it was not until more than a century later, in 1941, that it was again focused on by Kato [3]. Since then, pink teeth have been observed in burned corpses as well as after deaths due to poisoning, gunshots, freezing, natural causes and unknown causes [4,5,6,7,8,9]. At present, there is no consensus regarding the relationship between pink teeth and cause of death because the phenomenon appears to be affected by conditions such as humidity and the position of the corpse [2,6,8,10,11,12]. In addition, most research on the pink teeth phenomenon to date has been case reports and reviews based on observational studies, and there are very few experimental research reports on the mechanism underlying the pink teeth phenomenon.

According to Sakurada [13], Kato conducted a series of experimental studies related to the effects on oral tissues of asphyxiation caused by cervical compression and published four papers in classical Japanese in 1941 [3,14,15,16]. Those experimental studies using living rabbits examined changes immediately after asphyxiation, clarifying that the presence of pink teeth with little or no putrefaction might be indicative of death from asphyxiation caused by cervical compression. However, many cadavers that undergo a forensic autopsy may have putrefaction, and Kato did not perform a detailed examination of the change in color based on a consideration of the effects of putrefaction or the mechanism of its occurrence. Studies using living animals such as dogs, cats and rats have also been reported [17,18,19], but the experimental models may not be applicable to characterizing changes in color development over time in human teeth, given that they involved animals with tooth structures different from those of human teeth, and they also differed in terms of sample preparation methods and procedures, as well as tooth preservation conditions [4,19]. Furthermore, from an ethical point of view, it is now extremely difficult to conduct asphyxiation experiments using a large number of live animals. Although there have been some studies that used artificial capillaries as a model for tooth dentin experimental reports that used teeth extracted from cadavers [14,20,21,22,23], the conditions for observing pink teeth are not uniform, and the reproducibility and reliability of these approaches have not been verified. Thus, an experimental model for elucidating the pink teeth phenomenon has yet to be established.

Furthermore, no criteria have been published for classifying the color of pink teeth, so any tooth with a pinkish tone that can be discriminated from teeth with a normal color have been simply regarded as pink teeth [1,2,3,4,5,6,7,8,10,12,17,18,20,21,23,24,25,26,27,28,29,30,31,32,33,34]. Their evaluations are considered to depend on the subjectivity of the observer, and it is thus considered necessary to introduce a more objective color evaluation method.

In the present study, we used bovine teeth as an experimental model in order to clarify the mechanism by which pink teeth occur. They are similar in structure to human teeth but are larger and thus are often used as experimental materials in the field of dentistry. Studies in which bovine teeth have been used to simulate human teeth include the following: research on the effects of photopolymerization-type bonding agents, which are adhesives for composite resins used for tooth restoration [35]; research on the effects of collagen higher-order structures on adhesion to dentin [36]; and research on the effect of smear layer properties on adhesion to dentin [37]. Moreover, methods using hue band histograms or ΔE and the L*C*h color space can more objectively evaluate changes in color development and fading. In particular, the ΔE and L*C*h color space methods have been used in studies on post mortem color changes in the hair, skin, corneas, necrotic spots, etc. [38,39,40,41,42,43], and we speculate that they might also be useful for our experiments.

In the present study, we first used bovine teeth and blood to establish an experimental model of neck pressure being applied and then attempted to evaluate tooth color development and fading under various conditions (e.g., oxygen, light and temperature), using the ΔE and L*C*h color space methods.

## 2. Materials and Methods

### 2.1. Creation of the Experimental Model

Lower incisor teeth (n = 187) from 20-month-old cattle and bovine blood containing citric acid were obtained from Tokyo Shibaura Organ Co., Ltd. (Tokyo, Japan). With reference to a previous study [42], the teeth were immersed in hydrogen peroxide (Wako Pure Chemical Industries, Ltd., Osaka, Japan) at three different concentrations (2.5%, 7.5% and 15%), and the degree of periodontal ligament removal and the effect on color development were compared. In another study [21], blood was injected after pulpectomy; however, because pink teeth are observed in vital teeth, we compared color development between teeth with and without pulp. After cleaning and drying the crown and root, a syringe (SS-10S7P, Terumo Corp., Tokyo, Japan) was used to inject 400 μL of hemolyzed bovine blood through the apical foramen, and the state of color development was compared and examined in three directions: upward-facing, laterally and downward-facing. The control group was stored under the following conditions: 26 °C, 55% humidity, light-shielded and an anaerobic (nitrogen replacement) environment. The prepared pink tooth was placed on a blackout curtain, and after capturing images with a digital camera (E-M1 Mark2, Olympus Corp., Tokyo, Japan) equipped with a microlens (NY1S-MOFA, Olympus Corp.) and an iPhone SE (Apple, Inc., Cupertino, CA, USA) from a height of 11 cm, a comparative evaluation was performed. The enamel of bovine teeth is about twice as thick as that of humans, and the labial side of the cervical portion of the root was chosen as the evaluation site because it is where the pink coloration is most easily observed.

### 2.2. Color Evaluation Method

We compared two evaluation methods based on hue circle histograms, ΔE and the L*C*h color space. After photographing the prepared pink tooth, the image was imported to a personal computer, the RGB values were extracted using software at 504.0635 pixels/cm, and the hue band histogram, ΔE and L*C*h color space values were calculated. ΔE represents the difference in color tone, and in this study, ΔE > 20.0 was regarded as a color difference. The L*C*h color space represents lightness, chroma and hue angle and was used to evaluate color density, vividness and hue. A conversion formula is shown below.

R′G′B′ (corrected)
(1)$$R^{\prime }={\left(\left(R/255+0.055\right)/1.055\right)}^{2.4}$$
(2)$$G^{\prime }={\left(\left(G/255+0.055\right)/1.055\right)}^{2.4}$$
(3)$$B^{\prime }={\left(\left(B/255+0.055\right)/1.055\right)}^{2.4}$$

XYZ color system
(4)$$X=(0.4124R^{\prime }+0.3576G^{\prime }+0.1805B^{\prime })\times 100$$
(5)$$Y=\left(0.2126R^{\prime }+0.7152G^{\prime }+0.0722B^{\prime }\right)\times 100$$
(6)$$Z=\left(0.0193R^{\prime }+0.1192G^{\prime }+0.9505B^{\prime }\right)\times 100$$

CIELAB color system
(7)$$L^{*}={116\left(Y/100\right)}^{1/3}-16$$
(8)$$a^{*}=500\left\{{\left(X/98.072\right)}^{1/3}-{\left(Y/100\right)}^{1/3}\right\}$$
(9)$$b^{*}=200\left\{{\left(Y/100\right)}^{1/3}-{\left(Z/118.225\right)}^{1/3}\right\}$$

L*C*h color space
(10)$$C=\sqrt{{\left(a^{*}\right)}^{2}+{\left(b^{*}\right)}^{2}}$$
(11)$$h={tan}^{-1}\left(b^{*}/a^{*}\right)$$

Color difference
(12)$$∆E={\left\{{\left({∆L}^{*}\right)}^{2}+{\left({∆a}^{*}\right)}^{2}+{\left({∆b}^{*}\right)}^{2}\right\}}^{1/2}$$

### 2.3. Intra-Examiner and Inter-Examiner Reliability in Color Evaluation

Intraclass correlation coefficients (ICCs) were analyzed to confirm the reproducibility of the color evaluation method. ICCs are used to assess how well independent measurements such as differences between different devices and examiners correlate with each other. For the evaluation of intra-examiner reliability, under control conditions, the RGB values at the start, 7 days and 3 months were re-evaluated after 1 month, and the respective correlation coefficients were compared. The inter-examiner reliability evaluation was performed by one other evaluator using the same settings as those of the intra-examiner reliability evaluation for subjects, period and reference values, and the results were compared.

### 2.4. Comparison of Color Tone Due to External Environmental Factors

For the control conditions, time (6 h, 12 h, 1 day, 3 days, 7 days, 1 month and 3 months), oxygen (anaerobic and aerobic), light (shading and light), temperature (26 °C and 4 °C), humidity (55% and 90%) and soft tissue (absent and present) were comparatively evaluated. The change over time in the soft tissue condition was evaluated in the same way as that in the control condition.

### 2.5. Relationship with Carboxyhemoglobin and Hemoglobin Degradation Products

Pink tooth dentin prepared for 6 h to 3 months under control conditions was excised with a #80 dental reamer and dissolved in physiological saline in order to prepare a 20 mg/mL solution. After standing for 20 min, the samples were centrifuged at 6000 rpm for 20 min and measured at various absorbances, using Nano Drop One/One (Thermo Fisher, Waltham, MA, USA). A range of 538 nm [43,44,45] was used for carboxyhemoglobin, 383 nm [46] was used for total heme pigment, and 455–575 nm [47,48] was used for bilirubin.

### 2.6. Relationship with Red Pigment-Producing Bacteria

Cervical swabs of pink teeth prepared for 6 h to 3 months were used as the control group. DNA extraction was performed using the QIA amp DNA Mini Kit (Qiagen, Hilden, Germany) according to the manufacturer’s protocol. Briefly, samples (n = 5) were treated with 290 µL of buffer ATL and 10 µL of proteinase K (Thermo Fisher Scientific) at 56 °C for 15 min and eluted with 100 µL of Buffer AE to prepare the measurement samples. Thermal Cycler Dice^®^ Real Time System III (TaKaRa, Tokyo, Japan) was used for quantification. The genomic DNA of *Serratia marcescens subsp. marcescens* (RIKAKEN DNA Bank catalog JCM1239T) diluted 10^5^-fold with Nuclease-Free Water (Qiagen) was used as a positive control. Because it was difficult to collect an equal amount of the samples in this study, two types of primers, one with high specificity to *S. marcescens* and another with low specificity that reacts with all bacteria in the target, were used. The quantitative heterogeneity of samples was adjusted by assessing the proportion of *S. marcescens* in the total bacteria. The former primer of *S. marcescens* was as follows: forward (5′-GGTGAGCTTAATACGTTCATCAATTG-3′) and reverse (5′-GCAGTTCCCAGGTTGAGCC-3′) [49], and the latter primer was as follows: forward (5′-AAACTCAAAKGAATTGACGG-3′) and reverse (5′-CTCACRRCACGAGCTGAC-3′) [49,50]. Using SYBR^®^ Premix ExTaq™ (Tli RNaseH Plus, TaKaRa), 12.5 µL of TB Green Fast I, 9 µL of Nuclease-Free Water and 1 µL of each primer (final 0.2 µM) were added to 2 µL of template DNA. The cycling conditions were an initial single cycle for 10 min at 95 °C (for activation) followed by 35 cycles of two-temperature cycling, consisting of 15 s at 95 °C (for denaturation) and 30 s at 60 °C (for annealing and polymerization) [49]. A *t*-test and the Dunn–Bonferroni post hoc test were used for the statistical analysis of all experiments.

## 3. Results

### 3.1. Establishing the Experimental Model

To remove soft tissue such as periodontal ligament, 2.5%, 7.5% and 15% hydrogen peroxide was used (n = 5 each). Soft tissue remained at 2.5% but was easily removed at 7.5%, with little effect on color development. At 15%, a significant decrease in color development was observed, so 7.5% was considered to be an appropriate concentration. By comparing the degree of coloring with and without pulp, it was determined that sufficient color development could not be reproduced when pulp was present, and the ΔE value at 7 days was 11.14 ± 0.37 (n = 5) with pulp and 3.72 ± 3.72 (n = 5) without pulp. The ΔE value when pulp was absent was significant (*p* < 0.01). Based on these results, we decided to use teeth after pulpectomy in this study, as in previous studies [21]. Furthermore, regarding the state of tooth preservation, we observed the development of color in three directions, with the crown facing upward, laterally and downward. When the crown was facing downward, the color development was clear. However, when the crown was facing upward or laterally, color development tended to be weaker because of blood outflow. Therefore, teeth were stored in a downward-facing position. Images captured with the digital camera showed marked shadows on the margins, whereas those captured with the iPhone SE did not. A previous report [51] also showed the usefulness of the iPhone SE for the color tone evaluation, so we decided to use an iPhone SE in this study as well. The results are shown in Figure 1.

### 3.2. Color Evaluation Method and Evaluation of Intra- and Inter-Examiner Reliability

Of the color evaluation methods, the method using hue band histograms with evaluation items limited to hue only showed a change in the waveform over time (Figure 2), with significantly higher values at 6 and 12 h compared with 7 days (*p* < 0.01). In the method using ΔE (color difference) and the L*C*h color space, ΔE was significantly lower at 7 days (*p* < 0.01), compared with that at 6 and 12 h, and at 3 months, compared with that at 7 days (*p* < 0.05) (Figure 3a). L* (lightness) was significantly lower at 3 and 7 days than that at 6 h (*p* < 0.05); C* (chroma) was significantly higher at 7 days than that at 6 h (*p* < 0.01); and h (hue angle) was significantly lower at 7 days than that at 6 h, 12 h and 3 months (*p* < 0.01) (Figure 3b–d). These results suggest that ΔE and the L*C*h color space are more useful compared with hue band histograms.

Therefore, we attempted to evaluate the intra- and inter-examiner reliability of the method using ΔE and the L*C*h color space. The results are shown in Table 1. Regarding the intra-examiner reliability, the ICC of the RGB values was 0.95 or more at all time points, indicating high reproducibility (*p* < 0.01). In contrast, the inter-examiner reliability showed a low G value (ICC = 0.939182) at 3 months, although a high internal consistency (ICC = 0.95–0.99) was obtained for the other items (*p* < 0.01). Although the L*a*b* color system is the most widely used system for color tone evaluations along with ΔE, it can be used interchangeably with the L*C*h color space. Therefore, we decided to use ΔE and the L*C*h color space for the color tone evaluation in the detailed experiments that followed.

### 3.3. Color Change Due to External Environmental Factors

The results for various influencing factors (oxygen, light, temperature, humidity and soft tissue: n = 5) are shown. As shown in Figure 4, under aerobic conditions, L* (lightness) was significantly higher, and C* (chroma) was significantly lower from 1 day to 3 months (*p* < 0.05). In addition, h (hue angle) was significantly lower at 1 day and 3 months (*p* < 0.05). Under unshaded conditions, as shown in Figure 5, L* (brightness) was significantly higher at 7 days and 1 month (*p* < 0.05), and C* (chroma) was significantly higher at 1 day but significantly lower at 7 days (*p* < 0.05). In addition, h (hue angle) was significantly higher at 1 and 7 days (*p* < 0.05). The results for temperature, humidity and soft tissue are shown in Table 2. Under refrigeration, L* (lightness) showed significantly higher values at 3 days (*p* < 0.05); C* (chroma) showed significantly higher values at 1 and 3 days (*p* < 0.05); and h (hue angle) showed a significantly lower value at 3 days (*p* < 0.05). Under high humidity, L* (lightness) was significantly higher at 1 day, 3 days and 1 month (*p* < 0.05); C* (chroma) was significantly higher at 3 months (*p* < 0.05); and h (hue angle) showed a significantly higher value at 1 day but a significantly lower value at 3 days (*p* < 0.05). In the presence of soft tissue, L* (lightness) was significantly higher at 1 and 3 days (*p* < 0.05); C* (chroma) was significantly lower at 1 day (*p* < 0.05); and h (hue angle) showed a significantly lower value at 3 days (*p* < 0.05). In addition, as shown in Figure 6, in the transitional change under soft tissue conditions, L* (lightness) was significantly lower at 7 days and 1 month (*p* < 0.01), and C* (chroma) was significantly higher at 7 days (*p* < 0.01).

### 3.4. Relationship with Carboxyhemoglobin and Hemoglobin Degradation Products

As shown in Figure 7, the amount of CO-Hb in dentinal tubules showed a marked increase from 3 days and was significantly higher at 7 days than that at 6 h, 12 h and 3 months (*p* < 0.05). Although no significant difference was observed, the amount of total heme pigment showed an increasing tendency from 6 h to 1 day. The amount of bilirubin also showed a tendency to increase over time, although no significant difference was observed.

### 3.5. Examination of the Relationship with Red Pigment-Producing Bacteria

*S. marcescens* DNA was identified as having a unique melting peak at 87.6 ± 0.4 °C in the real-time PCR melting curve analysis, and DNA recovered from the cervical area of pink teeth also showed a peak at a similar position (Figure 8). As shown in Table 3, the amount of *S. marcescens* DNA in all bacteria showed an increasing tendency after 3 days and a significant increase at 1 month (*p* < 0.01). As shown in Table 4, regarding the ratio of *Serratia* to all bacteria, the amount of *Serratia* DNA showed a significant increase at 1 month (*p* < 0.01), although no significant difference was observed at other time points; however, at 1 and 3 days, the proportion of *Serratia* in the total bacteria decreased.

### 3.6. Limitations in Research

A limitation of this experimental model was that it did not fully consider the effects of other oral tissues to make blood into the congested state. Especially, because sufficient color development was reproduced when pulp was present, we used teeth after pulpectomy in this study, as in a previous report [21]. However, this method of removing pulp may not be considered as a result of various biomolecular and cellular exchanges that occur between the dental tissues and within the tubules. Therefore, it is essential to further improve this experimental model.

## 4. Discussion

Establishing an experimental model is important for clarifying the mechanism underlying the pink teeth phenomenon. Previous reports have used human, dog, cat and rat teeth or artificial capillaries [3,17,18,19,20,21,22,23]. However, there are large individual differences in human teeth. It can be difficult to collect a sufficient number of similar teeth for use in experiments, and the structure of animal teeth and artificial tubes differ substantially from that of human teeth. Thus, a more reliable experimental model is needed. In this study, we focused on bovine teeth, which are similar in structure to human teeth and are frequently used in the field of conservative dentistry. First, to establish the experimental conditions, bovine blood was injected at the root apex. To observe the change in color under conditions simulating a congested state, we examined the concentration of hydrogen peroxide that is necessary for removing soft tissues, the difference in color development depending on the presence or absence of dental pulp and the position for preserving teeth, with reference to previous studies. The results indicate that a 7.5% concentration of hydrogen peroxide was effective, and pulp extraction showed significant (*p* < 0.01) color development in the presence of pulp. These results are mostly consistent with a previous report [21]. In addition, we confirmed that, after injecting hemolyzed blood, a downward-facing position is best for storing the tooth, and the iPhone SE is useful for capturing images for later comparison. By observing the change in color over time under these conditions, we confirmed that color development started on the first day. A previous study [21] reported that the development of pink teeth began with hemolysis, and a pink teeth model using bovine teeth was considered to be useful for subsequent experiments. In addition, by comparing color evaluation methods, a red, pink and purple coloration—which is commonly recognized as pink teeth—is located near the minimum value of 0 and the maximum value of 255. From this characteristic, it was considered difficult to use the hue band histogram to evaluate pink teeth (Figure 2). In contrast, detailed information was obtained from the ΔE and L*C*h color space methods (Figure 3), with ΔE values being significantly higher at 7 days (the highest value was seen at 7 days in the control group), and thus it was considered that it might exhibit a color tone with a large tendency toward red. These results suggest that color development reached a peak at 7 days and then faded.

In terms of changes over time in L* (lightness), C* (chroma) and h (hue angle) under various conditions, significant differences were observed in aerobic, light and low-temperature conditions (Figure 4 and Figure 5, Table 2). Under aerobic conditions, L* (lightness) was high, and h (hue angle) was negative, indicating that the brightness of the pink coloration was low (pale) and tended to be blue. It was suggested that the aerobic condition might suppress the color development of pink teeth, given that blue is the complementary color of pink. Under light conditions, at 7 days, L* (lightness) and h (hue angle) increased, whereas C* (chroma) decreased, showing a less bright (pale) pink color with a reddish tinge. In other words, it was suggested that the light condition may have faded the pink color. Under low-temperature conditions, C* (chroma) was significantly higher at 1 and 3 days, indicating the possibility of developing a bright pink color under low-temperature conditions. Similar to the bright red color observed in mortal remains [43], it is suggested that the decomposition of hemoglobin is retarded under low-temperature conditions. In addition, the h (hue angle) of soft tissue as well as humidity, both of which are influencing factors, were significantly lower at 3 days, when a large red tendency was observed. Because 3 days is the time when decomposition begins [43], it was confirmed that decomposition had an effect on color development. Furthermore, as shown in Figure 6, the soft tissue condition showed the same temporal change as that of the control condition because L* (lightness) was significantly lower and C* (chroma) was significantly higher at 7 days than those at 1 day. However, no significant increase in L* (lightness) and h (hue angle) was observed at 7 days. This suggests the possibility that fading is slow under soft tissue conditions. Humidity factors are considered to be involved not only in putrefaction but also in capillarity in dentinal tubules [17].

Compared with other forms of hemoglobin, carboxyhemoglobin is said to be more stable and to have a characteristic bright red color [43]. As decomposition progresses, bacteria of the genera *Streptococcus* and *Proteus* proliferate, producing heme oxygenase to decompose the heme in the putrefactive product, thereby generating carbon monoxide [52]. Figure 7 shows that the change in carboxyhemoglobin content over time in this study was the highest at 7 days, showing the same change over time as that of the control group, which was the darkest and tended to be reddish at 7 days. Therefore, it was confirmed that carboxyhemoglobin affects the color development of pink teeth. In addition, although the total heme pigment and bilirubin levels did not show significant changes, a tendency to increase over time was confirmed. From these results, it is presumed that hemoglobin that entered the dentinal tubules was degraded.

Many of the reports on *S. marcescens*, a red pigment-producing bacterium, are studies on whether the fungus produces a pink color [12], and there are few reports comparing the amount of *S. marcescens*. In the present study, we investigated the relationship between the time course of *S. marcescens* in pink teeth and the color tone by comparing the amount of *S. marcescens* DNA in the test sites. As shown in Table 3 and Table 4, changes in the amount of DNA in *S. marcescens* showed a tendency to increase over time, with a significant increase (*p* < 0.01) at 1 month. However, given that the coloring faded after 7 days, it is unlikely that *S. marcescens* plays a substantial role in pink tooth coloring. A previous study also investigated whether *S. marcescens* affects pink tooth coloration under specific conditions [29]. Therefore, *S. marcescens* cannot be the main cause of pink tooth coloration, although it is possible that it is predisposed to grow under the same conditions that are conducive to the development of pink teeth.

In this study, it was experimentally clarified that a putrid environment greatly affects the development of pink teeth. This result suggests that it would be difficult to associate the development of pink teeth with the cause of death under putrefaction conditions. However, Kato’s experimental reports [3,14,15,16], which are based on rabbits, indicate that the presence of pink teeth at a stage when putrefaction has little or no effect might be a finding that is indicative of death from asphyxiation caused by cervical compression. Therefore, it may be worth continuing to examine the relationship between various case data relatively soon after death and the occurrence of pink teeth.

## Figures and Tables

**Figure 1 diagnostics-13-02699-f001:**
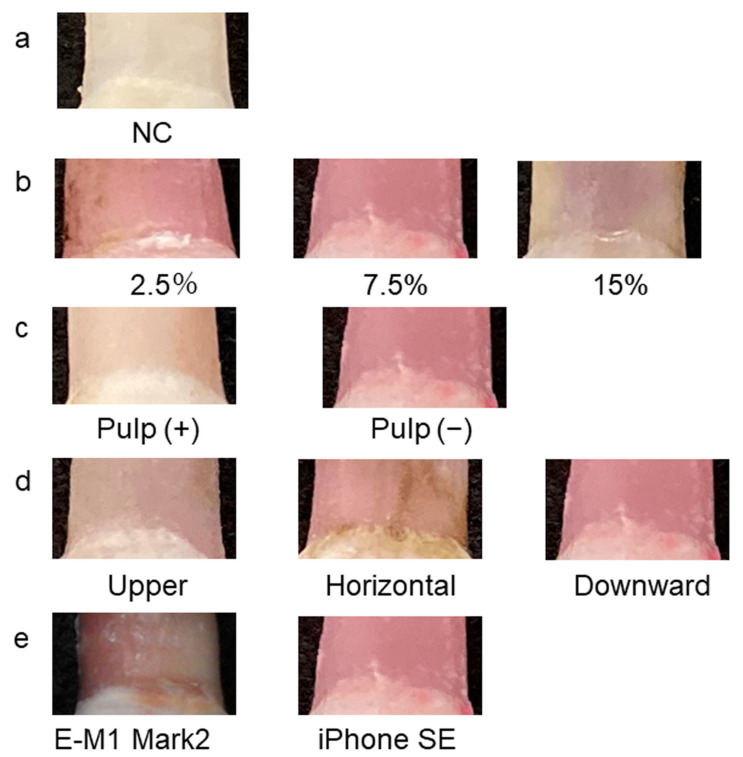
Pink teeth produced using the established experimental model. (**a**) Negative control (NC). (**b**) Examination of hydrogen peroxide concentration. (**c**) Examination of the presence or absence of dental pulp. (**d**) Examination of the position of the crown during the experiment; (**e**) Examination of shooting equipment.

**Figure 2 diagnostics-13-02699-f002:**
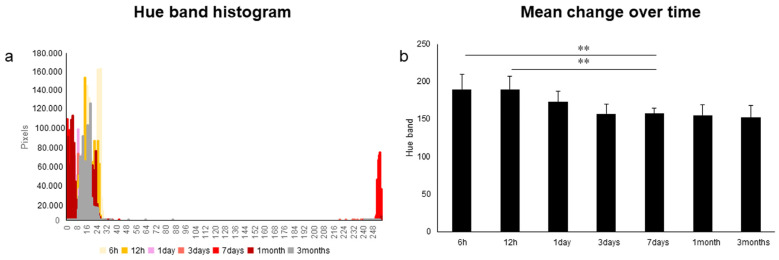
Evaluation using hue bands. (**a**) Hue band histogram. (**b**) Mean change over time. n = 5. ** *p* < 0.01.

**Figure 3 diagnostics-13-02699-f003:**
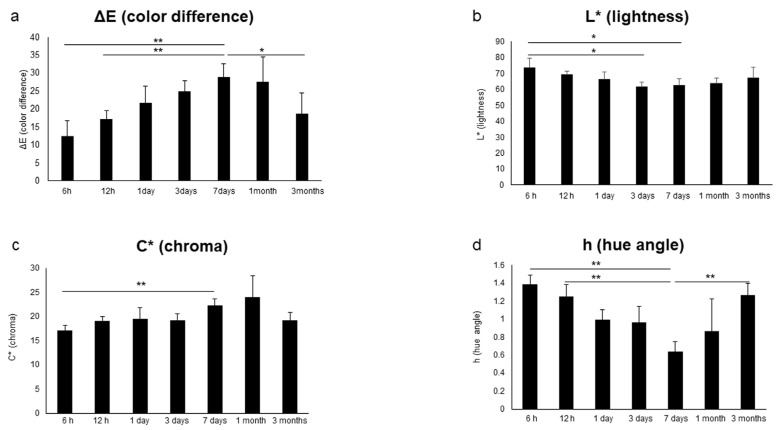
Methods using ΔE and L*C*h color space. (**a**) ΔE (color difference). (**b**) L* (lightness). (**c**) C* (chroma). (**d**) h (hue angle). n = 5. * *p* < 0.05, ** *p* < 0.01.

**Figure 4 diagnostics-13-02699-f004:**
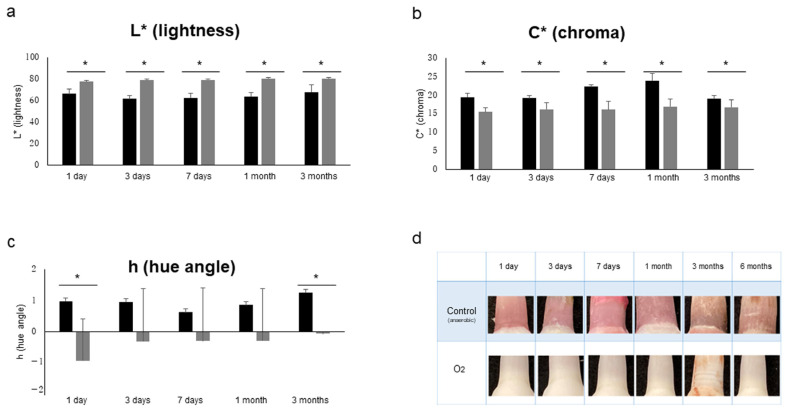
Effect of oxygen on color development. (**a**) L* (lightness). (**b**) C* (chroma). (**c**) h (hue angle). n = 5. * *p* < 0.05. Black bars represent control conditions (room temperature [26 °C], humidity [55%], shading and anaerobic conditions), and gray bars represent oxygen conditions. (**d**) Changes in pink teeth over time under control and oxygen conditions.

**Figure 5 diagnostics-13-02699-f005:**
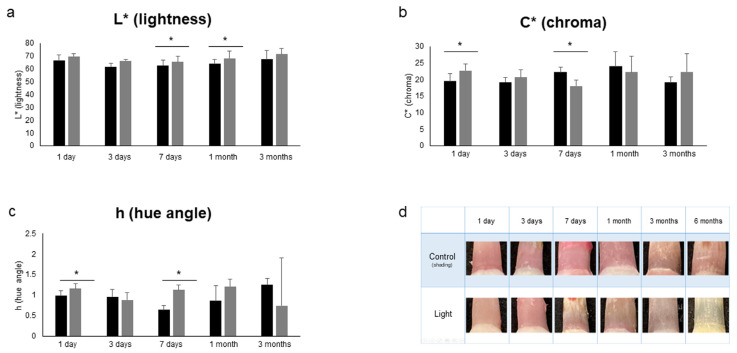
Effect of light on color development. (**a**) L* (lightness). (**b**) C* (chroma). (**c**) h (hue angle). n = 5. * *p* < 0.05. Black bars represent control conditions (room temperature [26 °C], humidity [55%], shading and anaerobic conditions), and gray bars represent light conditions. (**d**) Changes in pink teeth over time under control and light conditions.

**Figure 6 diagnostics-13-02699-f006:**
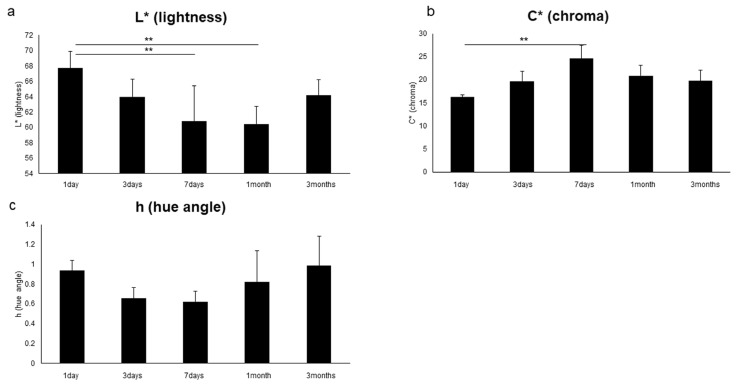
Effect of time and soft tissue on color development. (**a**) L* (lightness). (**b**) C* (chroma). (**c**) h (hue angle). n = 5. ** *p* < 0.01.

**Figure 7 diagnostics-13-02699-f007:**
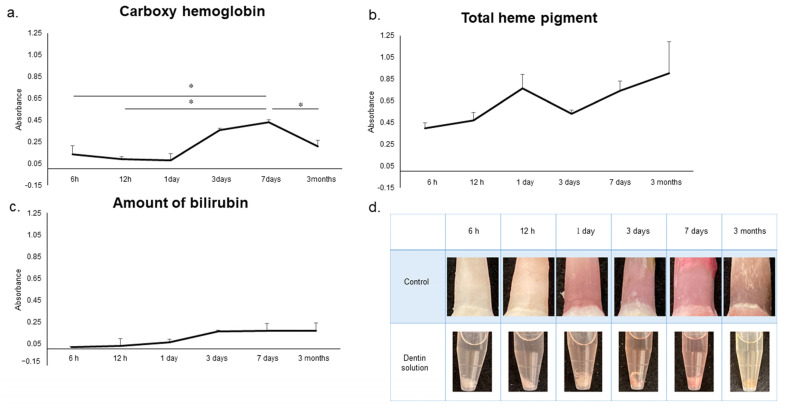
Relationship with hemoglobin and degradation products. (**a**) Carboxy hemoglobin. (**b**) Total heme pigment. (**c**) Amount of bilirubin. n = 5. * *p* < 0.05. (**d**) Changes over time in pink tooth and dentin solution under control conditions.

**Figure 8 diagnostics-13-02699-f008:**
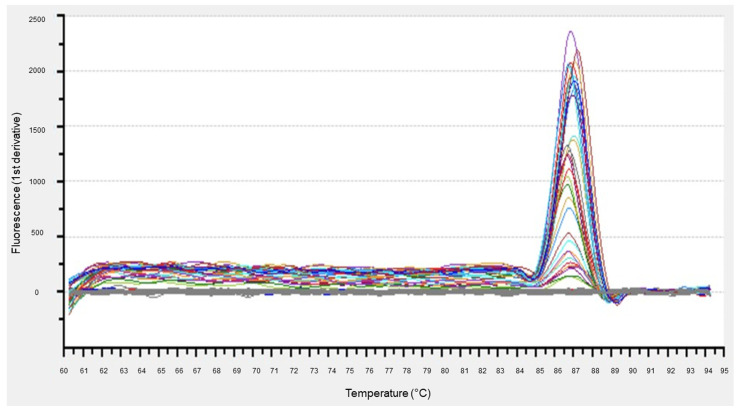
Real-time PCR melting curve analysis.

**Table 1 diagnostics-13-02699-t001:** Intraclass correlation coefficients for inter-rater and inter-device agreement (n = 5).

**Inter-Rater**	**R**	***p*-Value**	**G**	***p*-Value**	**B**
NC	0.997989	<0.01	0.997125	<0.01	0.997276
7 d	0.996241		0.997392		0.996532
3 m	0.994347		0.991968		0.993904
**Inter-Device**	**R**	***p*-Value**	**G**	***p*-Value**	**B**
NC	0.993376	<0.01	0.990016	<0.01	0.991847
7 d	0.993974		0.995711		0.995423
3 m	0.963627		0.939182		0.97046

NC, negative control; d, day; m, month.

**Table 2 diagnostics-13-02699-t002:** Comparison by various influencing factors.

	L*/Mean			C*/Mean			h/Mean		
	**Cont.**	**4 °C**	***p*-Value**	**Cont.**	**4 °C**	***p*-Value**	**Cont.**	**4 °C**	***p*-Value**
1 d	66.36	69.68	ns	19.52	23.00	<0.05	0.99	0.98	ns
3 d	61.67	65.73	<0.05	19.25	22.98	<0.05	0.96	0.82	<0.05
7 d	62.67	62.44	ns	22.48	20.85	ns	0.64	0.89	ns
1 m	63.87	61.95	ns	23.99	23.98	ns	0.86	0.82	ns
3 m	67.23	61.22	ns	18.90	19.80	ns	1.269	1.18	ns
	**Cont.**	**Humidity**	***p*-Value**	**Cont.**	**Humidity**	***p*-Value**	**Cont.**	**Humidity**	***p*-Value**
1 d	66.36	73.26	<0.05	19.52	18.05	ns	0.99	1.45	<0.05
3 d	61.67	68.40	<0.05	19.25	21.18	ns	0.96	0.84	<0.05
7 d	62.67	65.84	ns	22.48	24.44	ns	0.64	0.72	ns
1 m	63.87	68.33	<0.05	23.99	21.70	ns	0.86	1.26	ns
3 m	67.23	69.39	ns	18.90	24.43	<0.05	1.269	1.35	ns
	**Cont.**	**Soft tissue**	***p*-Value**	**Cont.**	**Soft tissue**	***p*-Value**	**Cont.**	**Soft tissue**	***p*-Value**
1 d	66.36	67.70	<0.05	19.52	16.22	<0.05	0.99	0.94	ns
3 d	61.67	63.97	<0.05	19.25	19.71	ns	0.96	0.66	<0.05
7 d	62.67	61.40	ns	22.48	25.02	ns	0.64	0.63	ns
1 m	63.87	60.43	ns	23.99	23.86	ns	0.86	0.82	Ns
3 m	67.23	64.20	ns	18.90	19.75	ns	1.269	0.99	Ns

Cont., control (room temperature [26 °C], humidity [55%], light-shielded and non-oxygen conditions); ns, no significant difference; d, day; m, month.

**Table 3 diagnostics-13-02699-t003:** Time course of *Serratia* DNA content.

Measurement Item	Qty (CP)/Mean	SD	*p*-Value
6 h	6.225 × 10^−6^	9.086 × 10^−6^	Ns
12 h	4.723 × 10^−6^	7.577 × 10^−6^	Ns
1 d	3.723 × 10^−6^	1.387 × 10^−6^	Ns
3 d	1.545 × 10^−5^	7.058 × 10^−5^	Ns
7 d	1.49 × 10^−5^	1.98 × 10^−5^	Ns
1 m	0.12361	0.0969252	<0.01

ns, no significant difference; h, hour; d, day; m, month. Dunn–Bonferroni post hoc test was used for statistical analysis.

**Table 4 diagnostics-13-02699-t004:** Changes over time in the percentage of *Serratia* DNA content in all bacteria.

Measurement Item	Ratio	CV	*p*-Value
6 h	0.023	3.95 × 10^−2^	Ns
12 h	0.025	3.03 × 10^−2^	Ns
1 d	0.00062	2.24 × 10^−1^	Ns
3 d	0.00060	1.18	Ns
7 d	5.50	3.40 × 10^−4^	Ns
1 m	1.15	8.26	<0.01

ns, no significant difference; h, hour; d, day; m, month. Dunn–Bonferroni post hoc test was used for statistical analysis.

## Data Availability

The data presented in this study are available on request from the corresponding author upon reasonable request.

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
