# Peer review of "Study on the Mechanism of the Pink Tooth Phenomenon Using Bovine Teeth: A Pilot Study"

_diagnostics, 2023, doi:10.3390/diagnostics13162699_

Round 1
Reviewer 1 Report
The authors carried out a model experimental study on the phenomenon of pink teeth coloration, starting from previous studies already conducted on the subject, trying to find a relationship between changes in teeth color tone and hemoglobin monoxide, its decomposition products and with bacteria producing red pigments, in the presence of various external environmental factors.
The phenomenon of "pink teeth" is nonspecific one and related to many, factors and variables, has already been widely described in the literature by several authors and is quite debated.
Their experimental model is carefully explained in the materials used, both on the use of bovine teeth and the evaluation of coloration.
The study seeks to find a reliable scientific methodology to be applied to the phenomenon of pink teeth (itself not easy to analyze) with objectivity in both data acquisition and evaluations made, and with fewer revisions to the text it can be considered for publication (e.g., parts in the introduction that are repeated in the discussion should be removed or the repetition of concepts in the text should be avoided - e.g. lines 81 , 94 and 304-305)
It would be fair to report the limits of in vitro research of the work, although it is well explained and made explicit why teeth after pulpectomy are used in the study: tooth vitality, unfortunately and in fact, is not obtained by putting blood back after emptying the pulpal cavity because it means not considering all the biomolecular and cellular exchanges that take place between the dental tissues and within the tubules.
Author Response
Response to Reviewers
Manuscript ID: diagnostics-2549344
Title: Study on the Mechanism of the Pink Tooth Phenomenon Using Bovine Teeth: A Pilot Study
Type: Article
Reply to the comments of Reviewer #1:
Thank you for reviewing our manuscript and providing valuable comments and suggestions. We have responded to your concerns and revised our manuscript as follows. Please find attached the revised version of our manuscript, with changes highlighted in green.
Comment #1
The authors carried out a model experimental study on the phenomenon of pink teeth coloration, starting from previous studies already conducted on the subject, trying to find a relationship between changes in teeth color tone and hemoglobin monoxide, its decomposition products and with bacteria producing red pigments, in the presence of various external environmental factors. The phenomenon of "pink teeth" is nonspecific one and related to many, factors and variables, has already been widely described in the literature by several authors and is quite debated. Their experimental model is carefully explained in the materials used, both on the use of bovine teeth and the evaluation of coloration.
The study seeks to find a reliable scientific methodology to be applied to the phenomenon of pink teeth (itself not easy to analyze) with objectivity in both data acquisition and evaluations made, and with fewer revisions to the text it can be considered for publication (e.g., parts in the introduction that are repeated in the discussion should be removed or the repetition of concepts in the text should be avoided - e.g. lines 81, 94 and 304-305)
It would be fair to report the limits of in vitro research of the work, although it is well explained and made explicit why teeth after pulpectomy are used in the study: tooth vitality, unfortunately and in fact, is not obtained by putting blood back after emptying the pulpal cavity because it means not considering all the biomolecular and cellular exchanges that take place between the dental tissues and within the tubules.
Response #1
According to the comments, we deleted lines 81, 94 and 304-305.
We agree with the comment “tooth vitality, unfortunately and in fact, is not obtained by putting blood back after emptying the pulpal cavity because it means not considering all the biomolecular and cellular exchanges that take place between the dental tissues and within the tubules”. We added the following sentence about a limitation of this experimental method, “We have fully understood that this in vitro experimental method by removing the dental pulp has never considered all the biomolecular and cellular exchanges that would take place between the dental tissues and within the tubules.”
In addition, we checked whether the cited references were relevant, and the references [52] was changed to [53] lines 350.
Reviewer 2 Report
After reviewing the manuscript “Study on the Mechanism of the Pink Tooth Phenomenon Using Bovine Teeth: A Pilot Study” according to the criteria for publication of the journal Diagnostics, we have made ​​the following assessment:
Review Criteria: Although there is a large literature on pink tooth phenomenon, therefore the study on the mechanism to establish a correct differential diagnosis (antemortem pink tooth, staining by saprophytic fungi in historical cases, i.e.) is of foremost importance, and in this field there are few published studies. Moreover, there is an almost total absence of work in relation to the changes in the color tone of postmortem pink tooth and the relationship with hemoglobin monoxide and its decomposition products or the changes with the proliferation of red pigment-producing bacteria. For this reason, this study is of maximum interest from the forensic point of view.
The clarity in drafting, and its approach is adequate. The review is exhaustive, and bibliography is representative of the ideas of the paper. We found the methodology of the study adequate both to establishing the experimental model and for control evaluation method.
Therefore, our proposal is to publish without modifications.
Author Response
Response to Reviewers
Manuscript ID: diagnostics-2549344
Title: Study on the Mechanism of the Pink Tooth Phenomenon Using Bovine Teeth: A Pilot Study
Type: Article
Reply to the comments of Reviewer #2:
Thank you for reviewing our manuscript and providing valuable comments and suggestions. We have responded to your concerns and revised our manuscript as follows.
Comment
After reviewing the manuscript “Study on the Mechanism of the Pink Tooth Phenomenon Using Bovine Teeth: A Pilot Study” according to the criteria for publication of the journal Diagnostics, we have made ​​the following assessment:
Review Criteria: Although there is a large literature on pink tooth phenomenon, therefore the study on the mechanism to establish a correct differential diagnosis (antemortem pink tooth, staining by saprophytic fungi in historical cases, i.e.) is of foremost importance, and in this field there are few published studies. Moreover, there is an almost total absence of work in relation to the changes in the color tone of postmortem pink tooth and the relationship with hemoglobin monoxide and its decomposition products or the changes with the proliferation of red pigment-producing bacteria. For this reason, this study is of maximum interest from the forensic point of view.
The clarity in drafting, and its approach is adequate. The review is exhaustive, and bibliography is representative of the ideas of the paper. We found the methodology of the study adequate both to establishing the experimental model and for control evaluation method.
Therefore, our proposal is to publish without modifications.
Response
Thank you very much for your kind comments.